# Application of Three-Dimensional Digital Photogrammetry to Quantify the Surface Roughness of Milk Powder

**DOI:** 10.3390/foods12050967

**Published:** 2023-02-24

**Authors:** Haohan Ding, David I. Wilson, Wei Yu, Brent R. Young, Xiaohui Cui

**Affiliations:** 1Science Center for Future Foods, Jiangnan University, Wuxi 214122, China; 2School of Artificial Intelligence and Computer Science, Jiangnan University, Wuxi 214122, China; 3Electrical and Electronic Engineering Department, Auckland University of Technology, Auckland 1010, New Zealand; 4Department of Chemical & Materials Engineering, University of Auckland, Auckland 1010, New Zealand; 5School of Cyber Science and Engineering, Wuhan University, Wuhan 430072, China

**Keywords:** 3D image analysis, surface roughness, milk powder, contour slice analysis

## Abstract

The surface appearance of milk powders is a crucial quality property since the roughness of the milk powder determines its functional properties, and especially the purchaser perception of the milk powder. Unfortunately, powder produced from similar spray dryers, or even the same dryer but in different seasons, produces powder with a wide variety of surface roughness. To date, professional panelists are used to quantify this subtle visual metric, which is time-consuming and subjective. Consequently, developing a fast, robust, and repeatable surface appearance classification method is essential. This study proposes a three-dimensional digital photogrammetry technique for quantifying the surface roughness of milk powders. A contour slice analysis and frequency analysis of the deviations were performed on the three-dimensional models to classify the surface roughness of milk powder samples. The result shows that the contours for smooth-surface samples are more circular than those for rough-surface samples, and the smooth-surface samples had a low standard deviation; thus, milk powder samples with the smoother surface have lower *Q* (the energy of the signal) values. Lastly, the performance of the nonlinear support vector machine (SVM) model demonstrated that the technique proposed in this study is a practicable alternative technique for classifying the surface roughness of milk powders.

## 1. Introduction

Quality control and product consistency are key properties for any food industry, and especially for the dairy industry [1]. Experience shows that the surface appearance usually influences consumers’ assumptions about the organoleptic and functional performance of milk powders, particularly for those consumers who use the product regularly and have become accustomed to the surface appearance of the product [2]. Additionally, it is inevitable for consumers to assess the quality of milk powders through visual perceptions of the product, so, consequently, variations in the surface appearance may lead to the customers considering that these milk powder products are counterfeit or even unsafe [3]. Therefore, the surface appearance is particularly important for milk powders, and visual consistency is necessary for milk powder plants. However, since different process conditions may affect the properties and appearance of milk powders [4,5,6], it is problematic for different plants to maintain the visual consistency of milk powders. In addition, the traditional way to grade the roughness of milk powders is to use sensory panelists which is empirical and laborious. Consequently, it is essential to find an efficient and reliable way to assess the surface roughness of milk powders and to maintain the visual consistency of milk powder products.

Computer vision techniques that have the ability to quantify the surface environments [7] were used by many studies to acquire texture features from various images [8,9,10], and were used to measure the surface roughness of components used in engineering [11,12,13,14]. Three-dimensional laser scanning methods that can generate geometrical triangulated data using a non-contact active method [15] have been used previously in the texture analysis of some photographs involving metal and concrete [16,17,18]. Alternatively, photogrammetry that can reconstruct three-dimensional (3D) models by stitching together numerous images from various positions [19] is a cheaper alternative to 3D laser scanning and has become a practicable method in 3D reconstructions or surface texture analysis [20,21,22,23,24]. However, many studies used photogrammetry to analyze the texture of soil and sediments [25,26,27]. Moret-Fernández et al. [28] utilized photogrammetry to evaluate the bulk density of small soil aggregates, while Merel and Farres [29] also used photogrammetry to measure the surface evolution and microrelief caused by erosion, and found that this method is sufficiently accurate. In addition, Moret-Fernández et al. [28] stated that it is better for the novice to use automated software to process photogrammetry, and the time required for processing data will be higher if the system is manual [30,31,32]. Furthermore, the fast Fourier transform (FFT) spectra were used in many studies to extract the textural feature from images [33,34,35].

To date, standard laboratory tests, such as flowability, water activity, bulk density, and particle size distribution, have been used to measure the functional properties of milk powders [36,37]. For example, a kinetic pulse nuclear magnetic resonance (NMR) technique that can measure the rehydration of milk powders was proposed in [38], Nijdam and Langrish [39] calculated the bulk density by computing the volume variation of milk powders in a graduated cylinder after tapping, while Lee et al. [40] measured the dispersibility, as well as the wettability of milk powders by calculating the variance between the electrical resistance of water and the electrical resistance of air. However, since these instrumental measures are different from human perception, these instrument measures may misjudge the sensory quality of products [3,41]. Additionally, the sensory quality of products has been analyzed by many texture analysis techniques. For instance, the sensory and texture properties of cholesterol-removed and whole milk cream cheese had been compared by Jeon et al. [42] throughout four weeks of storage, Gosselin et al. [43] analyzed the texture of polymer powders by using the GLCM technique, while Lille et al. [44] evaluated the flavor and appearance of snacks made from wholegrain rye flour and whole milk powder by using the sensory analysis. In addition, particle texture analysis (PTA), powder electron diffraction, and scanning electron micrographs are utilized to assess the surface of various products [45,46,47]. Furthermore, Traill et al. [3] utilized a trained sensory panel and a Rate-All-That-Applies technique to distinguish the roughness of milk powders, and determined that the significant distinction between milk powder samples is the size of milk powder lumps. Thus, Traill et al. [2] designed a photographic standard that can be utilized by grading assessors to classify commercial dairy powders into different lumpiness (roughness) groups, and demonstrated that this technique can grade the milk powders according to the level of visual lumpiness.

However, using the sensory panel to categorize the appearance of milk powders is subjective. Additionally, all the human evaluators need to be trained. In a previous related study, Ding et al. [48] proposed a purely geometric algorithm based on the 3D mesh which computes the area of the triangle formed by the three adjacent surface normals to classify the local surface smoothness of the milk powder samples. This study is based on the previous study [48], and the aim of the previous study is to explore the feasibility of classifying the visual appearances of different milk powders by using photogrammetry, while the aim of this continuing study is to improve on this single local method, to propose 3D image processing quantification algorithms, and to explore the reliability and feasibility of using this method to grade the surface appearance of different milk powder samples. In addition, the surface normal analysis which compares the area of triangle formed by the three adjacent surface normals as well as the angle between the adjacent surface normals was used in the previous study [48], and principal component analysis (PCA) was used to reduce the set of variables for the classifier, while a third-order polynomial nonlinear support vector machine (SVM) classifier was developed to classify the surface smoothness of milk powder samples in the previous study [48]. However, this work introduces a strategy based on analyzing contours and additionally, performing a frequency analysis of these curves to extract the high frequency components which are related to the lumpiness of the sample, and a second-order polynomial nonlinear support vector machine (SVM) model was chosen to grade the samples. Consequently, the methods used in this work are entirely different from the methods used in the previous work. Furthermore, the results in the previous work [48] showed that the surface normal analysis is effective for quantifying the surface appearance of milk powders, while it is expected that the results in this study will prove that the 3D digital photogrammetry techniques proposed in this work can effectively distinguish the visual appearance of milk powders and is a practicable alternative technique for classifying the surface roughness of milk powders.

## 2. Materials and Methods

### 2.1. Milk Powder Samples

The method of preparing milk powder samples in this study is the same as the method used in a previous study [48]. To duplicate the experience of a representative customer, the Fonterra Co-operative milk powders used were purchased off the shelf from a local Auckland, NZ, superstore. These milk powders comprise both instant whole and instant trim milk powders which allows the investigation of the effects of moisture level on the surface texture of various types of milk powders. The milk powders bought have similar moisture levels, though the surface texture properties of the milk powder samples may be different due to different moisture levels [48]. Therefore, to artificially create milk powders purchased at different locations and seasons, with corresponding differing moisture levels and surface appearances, varying amounts of water were sprayed on the samples. To assess the actual moisture level of each milk powder sample, three replicates of samples were dried in an oven (Cole-Parmer, Vernon Hills, IL, USA) [49,50] after photogrammetry tests, where the mass difference before and after drying is the weight of moisture contained, and the mean of three values was used. Four moisture levels were manually made in both samples (instant whole and instant trim milk powders), and these surface roughness grades are denoted as original, smooth, medium, and rough, where the first surface roughness grade (original) are the original milk powders. Three replicates of analysis were made for each moisture class for repeatability. Figure 1 shows the calibration chart with a somewhat arbitrarily chosen background image [48] which is subsequently in the image processing algorithm to robustly differentiate specific reference points. The squares of different colors in Figure 1 are used to distinguish the location of the model when building 3D models of milk powder samples.

### 2.2. Sample Preparation

As noted in Ding [48], it is important for the subsequent geometrically based analysis strategy that the milk powder cone is constructed in a consistent manner. Figure 2a shows the milk powder delivery device which contains a funnel with a stopper, a box, and a sample holder [48]. To maintain consistency in the sample preparation, the size of the funnel and the relative position of the funnel to the sample holder are the same. Various sizes of the funnel can make milk powder cones with different shapes and sizes. For each sample, 80 g of milk powders was released onto the sample holder below the funnel. The shape of all the milk powder cones is similar in order to increase the comparability. Additionally, Figure 2b shows the diagram of the photogrammetry equipment [48]. A Nikon D810 camera (Nikon, Tokyo, Japan), was used for the image capture and was fixed on a tripod. The distance from the camera to the samples was constant. In addition, to emphasize the boundary of the milk powder cones, and to eliminate the effects of excess shadow on the milk powder cones, four floodlights were turned on and placed on both sides of the samples. A black backing cardboard was used to maintain a constant background and lighting. Furthermore, over 60% overlap between spatially continuous photos is needed [51,52] for photogrammetry to develop 3D digital models of milk powder cones. Therefore, the milk powder cones on the turntable were photographed about every 11°, so that each sample has 33 images. Detailed figures of the milk powder delivery device and photogrammetry equipment are shown in Ding et al. [48], and an example picture of a cone is presented in Figure 3.

### 2.3. 3D Digital Models Building

The computer used for building 3D digital models was a Lenovo computer (Lenovo, Beijing, China), with an Intel (R) Core (TM) i9-10900 CPU @ 2.80 GHz with 32 GB installed RAM, and the graphics card of the computer is NVIDIA GeForce RTX 3060 (12 GB). 

After achieving all the photos of milk powder samples from photogrammetry equipment, the 33 images of each milk powder sample were separately imported into the software *RealityCapture* 1.0 (EPIC GAMES, Carrytown, NC, United States), [53,54] which builds the 3D triangular mesh of milk powder cones. Subsequently, the whole three-dimensional reconstructions with the texture of milk powder cones were created. The 3D triangular meshes were processed in *Matlab R2019b* (MathWorks, Natick, MA, USA). 

## 3. Three-Dimensional Image Analysis

In previous work, Ding et al. [48] built the surface normals of each triangle mesh for milk powder samples, and used the differences (area and angle formed by the adjacent surface normals) between the 3D milk powder models to measure the surface smoothness of milk powder samples. However, this work aims to slice the 3D milk powder samples into equally spaced contours, and use the frequency response of the deviations for each contour to quantify the surface roughness of milk powder samples, which is completely different from the method used in the previous study. To achieve this, we first extract and unwarp the contours from the cone models, then compute the differences between the unwrapped contours and the best-fit perfect circles. Finally, we can then analyze the roughness of milk powder samples by comparing the dominant frequencies of the wavelengths of expected lumpiness.

### 3.1. Contour Slice Analysis

In order to make the thickness between each layer as small as possible and to ensure that the data between each layer are enough for the subsequent analysis, around 40 equally spaced contours from the bottom of the cone to the top of the cone were used. The first step is to extract the contours of the cone, as shown in Figure 4. The contours will be circles if the milk powder sample is perfectly smooth and fall in a cone-shaped heap, while the roughness (lumpiness) of the sample is indicated by the deviations from the circles.

### 3.2. Frequency Analysis of the Deviations

For each contour, this technique was used to measure the features (dominant frequencies and shape) of the differences from circularity, and it is assumed that the lumpiness will deform the circularity of the contour. For each contour, the first step is to extract the boundary (*x*, *y*) at the given height. For an ideal cone, the extracted boundary will be a circle, but for the milk powder cone, the extracted boundary will diverge to some extent from a circle. Subsequently, the data of the contour was used to fit the least-squares circle, and the radius (*R*) of this circle was calculated. Given the true circle radius, the difference between the contour radius (*r*) and *R* as a function of angle around the circle can be calculated by Equation (1):(1)e(θ)=R−r(θ)

In order to take advantage of the efficient FFT, this trend will need to be interpolated on an equal grid spacing. The dominant frequencies (or alternatively the dominant wavelengths) and the variance of the difference between the unwrapped contour and the least-squares fit mean can be calculated if the deviations (at each altitude) are given. To better focus on the wavelength regions, the λ-axis can be abridged because the estimated sizes of the lumpiness are known. In addition, the frequency component is a spatial frequency (typically denoted β) since the basic measurement is a distance. Generally, longer data sets are needed to acquire a better resolution at the higher wavelengths, and it is necessary to sample at a finer resolution to better measure the higher frequencies. However, in this study, the finer resolution is not essential to better quantify the lumpiness, and furthermore, since the cone is circular, it is inappropriate to simply sample more data since one will then traverse again around the cone a second time. Lastly, the energy of the signal can be calculated by Equation (2): (2)Q=∫|FFT(e)|dλ

### 3.3. Support Vector Machine (SVM)

The SVM is a kernel-based method that is widely used to address pattern recognition problems [55] and binary classification problems [56]. To prevent overfitting, cross validation was used in this study. The data were separated into four subsets (each subset has 25% of the data). After trying different classifiers, a second-order polynomial nonlinear SVM classifier outperformed the others. Therefore, this SVM model was chosen to categorize the surface roughness of the samples. In addition, the confusion matrix [57], which is a classification assessment method [58], was utilized to assess the performance of the SVM classifier. The performance indicators, including specificity, overall accuracy, and sensitivity, are computed, and the detailed definition of these indicators is described in [59,60].

## 4. Results and Discussion

### 4.1. Milk Powder Cones

Table 1 shows the mean moisture values of all the moisture levels with standard deviation [48], while Figure 5 presents the front views of each sample [48]. Traill et al. [2] used photo standards to grade the surface roughness of milk powders by a trained sensory panel. Compared with the photo standards of lumpiness grades classified in [2], the original (first class) milk powder samples have a similar surface appearance to the level 0 dairy powders, and the rough (fourth class) milk powder samples have a similar appearance to the extreme appearance (level 14) of milk powders. Consequently, the original and rough milk powder samples are, respectively, referred to as Class 0 and Class 3. In addition, the smooth (second class) samples and the medium (third class) samples have a similar surface appearance to the moderate clumping samples (level 4–9) and the high clumping samples (level 9–13), separately. Thus, the smooth milk powder samples and the medium milk powder samples are separately denoted as Class 1 and Class 2.

From the figures, since the oblique vertical views of samples are clearly visually very different from each other, it is reasonable to assume that there is a strong correlation between the surface texture properties and the moisture level of powder samples. However, it is vital to discover a mathematical and robust method to quantify these relations. For each moisture level, three milk powder cones were fabricated to ensure the repeatability of the method.

Figure 6 shows an example 3D reconstruction triangular mesh model which has around 500,000 triangles of a sample. From Figure 6, it is notable that if the milk powder sample has no lumpiness (perfectly smooth), the milk powder cone will be a near-perfect circular-based cone, which means that the top view of the contour extracted from the three-dimensional milk powder model will be a perfectly concentric circle. On the other hand, the contours will show fluctuations around the perfect circles if the sample’s surface is rough (with some lumpiness).

Figure 7a shows the unwrapped contour of a Class 0 milk powder sample at the middle layer (layer 20), while Figure 7b presents an unwrapped contour of the Class 3 milk powder sample at the middle layer (layer 20). The red circles in the figures are the least-squares circles fitted by the data of the 3D digital models in this layer. From these trends, it is clear that the contour of the smooth milk powder sample shows a near-perfect circle, and the unwrapped contour of the smooth sample shows a correspondingly near-straight horizontal line. Conversely, the contour of the Class 3 sample appears irregular, and the unwrapped contour of the rough milk powder sample exhibits considerable fluctuations.

### 4.2. Contour Slice Analysis

The 3D digital model top views of Class 0–3 instant whole and trim milk powder samples are shown in Figure 8. It is obvious that the contours of the Class 0 milk powder samples are relatively circular showing that the Class 0 samples exhibit little lumpiness, while the contours of the Class 3 samples presented in Figure 8 are relatively irregular showing that there are many lumps in the Class 3 milk powder samples. Furthermore, although the Class 0 instant whole milk powder sample does not show any obvious lumpiness, on the whole, the surface of this sample is rougher than the surface of the Class 0 instant trim milk powder sample.

All contours (40 layers) for a Class 0 sample and a Class 3 sample are separately shown in Figure 9a,b. Note that the radius of the circle becomes increasingly smaller from the bottom to the top, which means that the bottom contour slices have more data than the top contour slices. Additionally, from Figure 9b, it is clear that there are more lumps on the bottom contour slices, which proves that the lumpiness tends to fall down the milk powder cone. It is also notable that all the contours of the Class 0 sample are more circular than the contours of the Class 3 sample, illustrating that the smoother the sample, the more circular the contours.

### 4.3. Variance of the Contours

The unwrapped contours of the Class 0–3 instant whole and trim milk powder samples are shown in Figure 10. It is worth mentioning that the higher the cone (the smaller of the radius), the fewer the data points of the contours. Additionally, the contours extracted from the smooth samples have fewer oscillations than the contours extracted from the rough samples, and the large oscillations represent the lumpiness. From Figure 10, it is clear that the Class 3 milk powder cones have the most lumps, while the Class 0 milk powder cones have almost no lumps. In addition, the Class 1 milk powder cones have a small number of lumps and the Class 2 milk powder cones slightly more lumps than the Class 1 milk powder cones. Furthermore, all the oscillations are concentrated in the contour slices with a small radius (bottom).

The deviations between the contours and the least-squares circles for each layer of Class 0–3 instant whole and trim samples are presented as a linear plot in Figure 11, and Figure 11 also plots the standard deviation for each altitude. A rough surface will have a higher standard deviation while the low standard deviation will be shown in the smooth surface, which is obvious in both the rough and smooth samples. For example, the standard deviations of the Class 0 cones are the lowest, while the standard deviations of the Class 3 cones are the highest. In addition, the Class 1 milk powder cones have a larger standard deviation than the Class 0 cones but a smaller standard deviation than the Class 2 milk powder cones. Additionally, the trends of the standard deviations in each Class milk powder cones are similar. Furthermore, since the lumpiness tends to fall down the milk powder cone, a higher variance is shown in the longer contours nearer the bottom.

### 4.4. Comparing the Frequency Responses

Figure 12a,b separately show the results of the contours, deviation, and frequency response of a Class 0 and Class 3 samples. For each sample, the absolute value of the frequency response was calculated, and the log of the wavelength against the log of the frequency response (which is easier to interpret) was presented in the figures. The linear perimeter distances for each layer (in the deviation plot) are simply computed by using the nominal radius (the following numbers) and the angle around the cone, and it is clear that the computed values are the approximation of the true distances. The independent scale in frequency response plots is shown in wavelength, and measured in distance units which are referred to as ‘meters’ for convenience. In addition, the DC component of the signal should be near zero because the mean value was subtracted. Consequently, the magnitude of the FFT is not adjusted at the Nyquist or DC frequency. Additionally, it is obvious that the unwrapped contours are comparatively flat for the smooth milk powder cone (Class 0 milk powder sample), but there is slight fluctuation for the shorter contours (higher layers). The top of the frequency response in Figure 12a also strengthens the assumption. However, the unwrapped contour trends are far more variable for the rough (Class 3) milk powder cone in Figure 12b. It is expected that the lumpiness might drop to the bottom of the milk powder sample. However, the deviation plot in Figure 12b tends not to demonstrate this.

Figure 13a,b separately compare the contours, deviations, and frequency response of Class 0–3 instant whole and trim milk powder samples at layer 20 (approximately the middle of the cone). To clearly show the shape of each contour, the radii of Class 1–3 are enlarged appropriately (the radii of these four samples are very similar) in the contour plots of Figure 13. Additionally, the axis is truncated in the frequency response plots of Figure 13 since the high frequency (low wavelength) has little useable information. From Figure 13a,b, the Class 3 milk powder sample shows more energy at wavelengths (λ) around 10 distance units, and the milk powder samples with the rougher surface show more energy at the higher wavelengths, which is expected. Furthermore, Figure 14a,b, respectively, compare the frequency response of Class 0–3 samples with a linear scale because the differences in frequency response are reduced due to the logarithmic scale in Figure 13a,b. The *Q* values (the energy of the signal) of each sample in Figure 14a,b are computed by Equation (2), and as expected, the milk powder samples with smoother surfaces have lower *Q* values. This result proves that it is appropriate to use *Q* values as the tool to categorize the roughness of the milk powder samples.

### 4.5. Classification of the Surface Roughness

Since the *Q* value of each layer can be calculated, the *Q* values of 40 layers for each milk powder sample were used to build the classifier to categorize the surface roughness of milk powders. Additionally, since each moisture level has 6 milk power cones (three instant trim cones and three instant whole cones), this study has 24 milk powder cones in total. Figure 15 presents the results of the nonlinear SVM classifier. The blue parts in the confusion chart represent the true positive and true negative predictions, and the false negatives and false positives from the predictions are shown as reddish cells. Table 2 lists the specificities and sensitivities of the developed SVM model. For the performance of the model developed in [53], only one milk powder sample was wrongly predicted, while from these results, all the smooth milk powder samples (Class 0) and the extremely rough-surface milk powder samples (Class 3) were predicted correctly. However, one Class 1 sample was incorrectly classified as a Class 0 sample, and two Class 2 milk powder samples were incorrectly classified as a Class 0 and a Class 1 sample, respectively. This may be because the lumps (roughness) in these three samples were not too distinct. Additionally, the overall accuracy of this classifier is 87.5% which is close to the overall accuracy of the classifier developed in [48] (the overall accuracy is about 88%). Furthermore, since the sample size (only eight samples) of the classifier developed in [48] is smaller than the sample size (24) of the classifier developed in this study, the reliability of this classifier is better than the reliability of the classifier developed in [48]. In addition, the loss (mean squared error) obtained by the cross-validated regression model is about 0.17, so that the accuracy (1—loss) of the cross validation is approximately 0.83, which means that this classifier performs well in the four-fold cross validation. Consequently, it is feasible to use this technique as a preliminary means to classify the milk powders into various surface roughness grades.

## 5. Conclusions

This study investigated the application of three-dimensional digital photogrammetry to classify the surface roughness of milk powder. The technique proposed in this study is objective and is an alternative to the traditional manual surface roughness classification methods. Different from the 3D image analysis methods used in [48], which classify the surface smoothness of milk powder samples by comparing the area of triangles formed by the three adjacent surface normals as well as the angles between the adjacent surface normals, the 3D digital photogrammetry techniques proposed in this study were used to classify the surface roughness of milk powder samples by comparing the variances and frequency responses of each contour slices between different milk powder samples. However, this research only considered four surface roughness classes. To improve the robustness of the classifier, more milk powder samples and surface roughness classes are recommended. From the results of the method proposed in this study, a higher standard deviation was observed on the rough surface, while the smooth surface had a low standard deviation. Furthermore, the milk powder samples with rougher surfaces had higher *Q* values (the energy of the signal), while the milk powder samples with smoother surfaces had lower *Q* values. Finally, the performance of the nonlinear SVM classifier demonstrated that the 3D image processing technique developed is a practicable alternative technique for classifying the surface roughness of milk powders.

## Figures and Tables

**Figure 1 foods-12-00967-f001:**
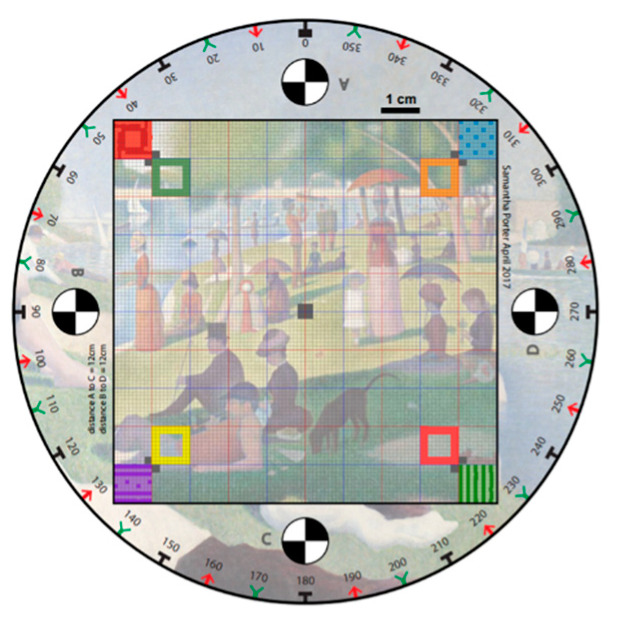
A calibration chart used to differentiate specific reference points.

**Figure 2 foods-12-00967-f002:**
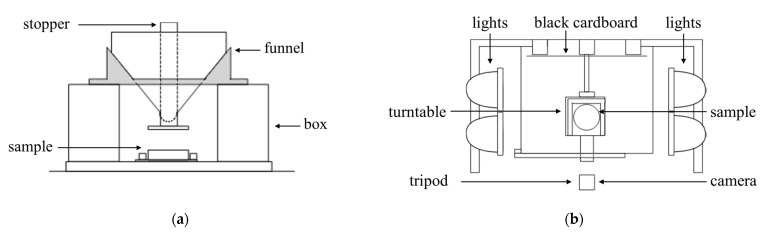
(**a**) The diagram of the milk powder delivery device; (**b**) A plan view of the photogrammetry equipment and sample stage.

**Figure 3 foods-12-00967-f003:**
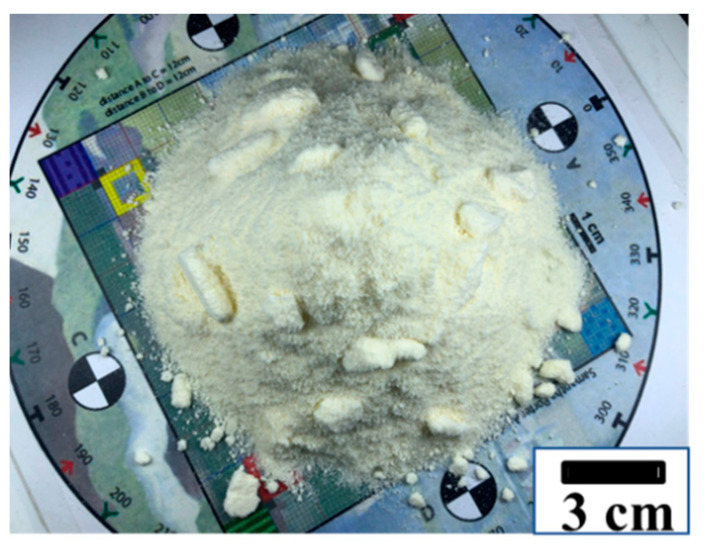
An example picture of a cone.

**Figure 4 foods-12-00967-f004:**
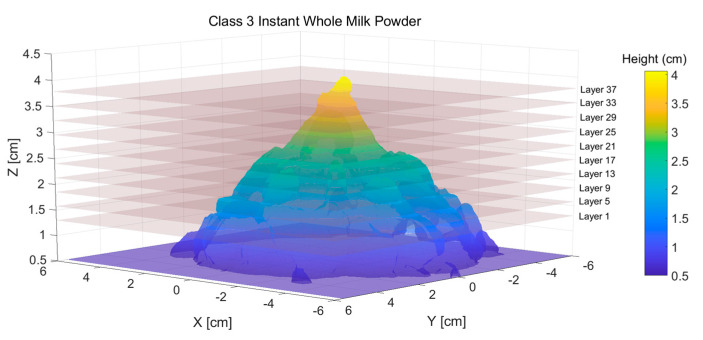
An oblique view of the triangulated model of a milk powder cone sliced at 40 equally spaced contours (only 10 slices are diagrammatically shown in this figure for clarity).

**Figure 5 foods-12-00967-f005:**
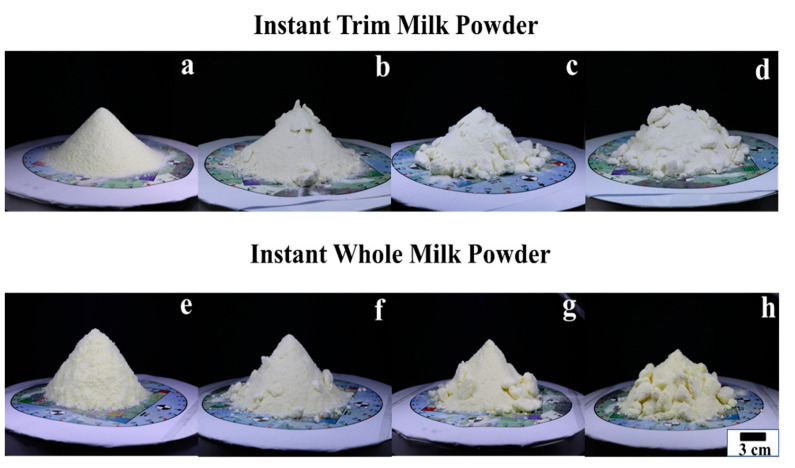
The front views of Class 0 (**a**), Class 1 (**b**), Class 2 (**c**), Class3 (**d**) instant trim milk powder samples and Class 0 (**e**), Class 1 (**f**), Class 2 (**g**), Class 3 (**h**) instant whole milk powder samples.

**Figure 6 foods-12-00967-f006:**
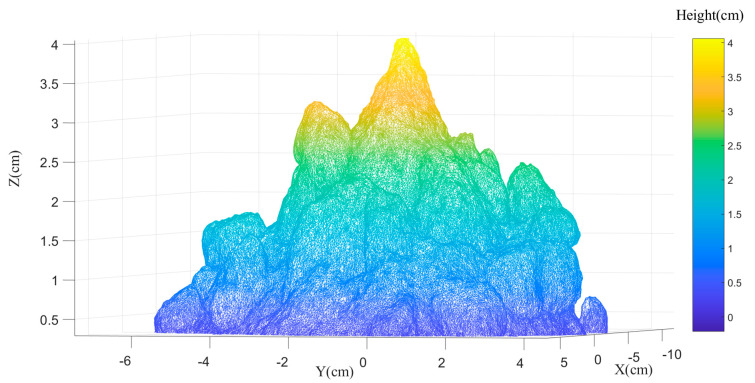
An 3D reconstruction triangular mesh model of a milk powder sample.

**Figure 7 foods-12-00967-f007:**
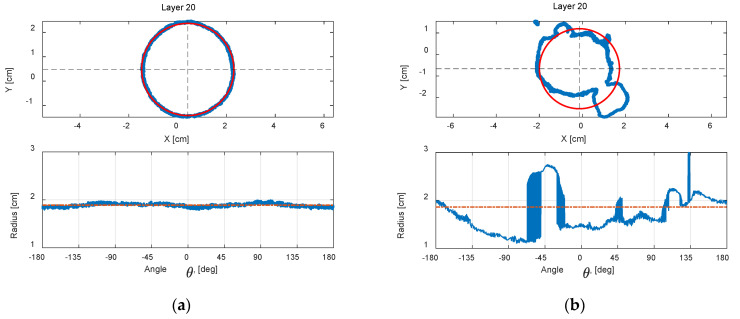
(**a**) An unwrapped contour of Class 0 milk powder sample at layer 20; (**b**) an unwrapped contour of Class 3 milk powder sample at layer 20.

**Figure 8 foods-12-00967-f008:**
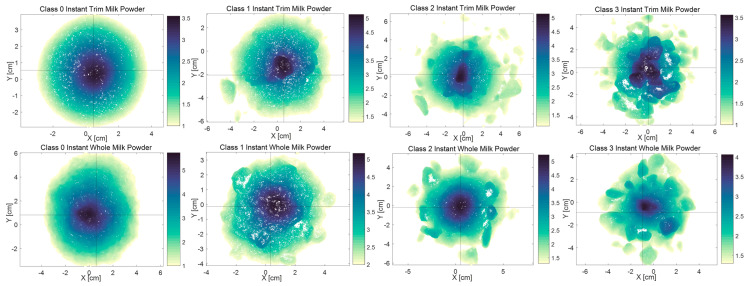
The 3D digital model top views of Class 0–3 milk powder samples.

**Figure 9 foods-12-00967-f009:**
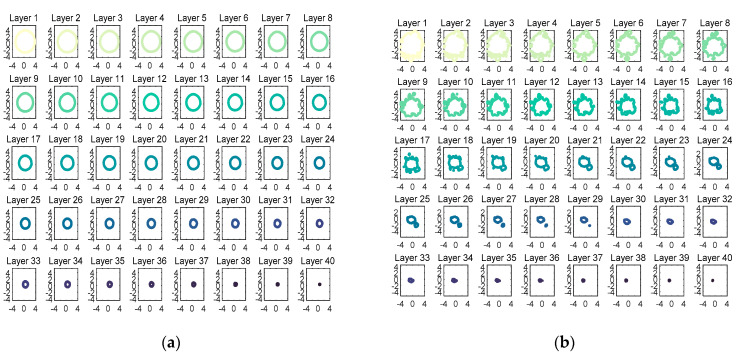
(**a**) All contours for a Class 0 milk powder sample; (**b**) all contours for a Class 3 milk powder sample.

**Figure 10 foods-12-00967-f010:**
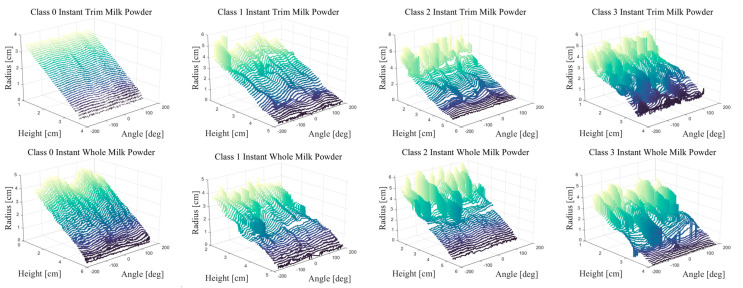
The unwrapped contours of Class 0–3 milk powder samples.

**Figure 11 foods-12-00967-f011:**
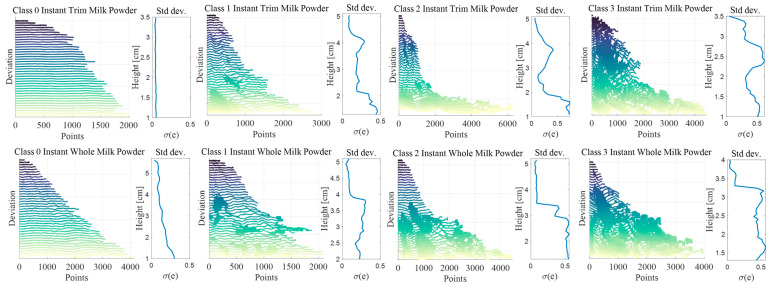
The deviations and the standard deviation of Class 0–3 instant whole and trim milk powder samples.

**Figure 12 foods-12-00967-f012:**
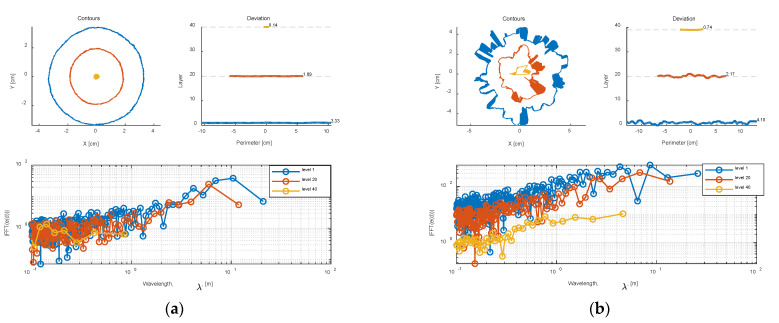
(**a**) The contours, deviations, and frequency response of a Class 0 sample; (**b**) the contours, deviations, and frequency response of a Class 3 sample.

**Figure 13 foods-12-00967-f013:**
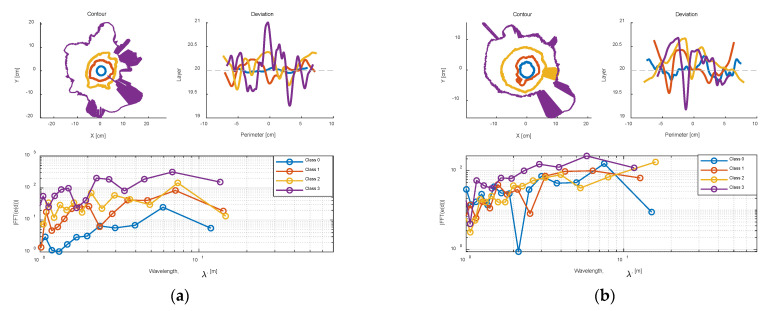
(**a**) Comparing the contours, deviations, and frequency response at layer 20 across the Class 0–3 instant trim samples; (**b**) comparing the contours, deviations, and frequency response at layer 20 across the Class 0–3 instant whole samples.

**Figure 14 foods-12-00967-f014:**
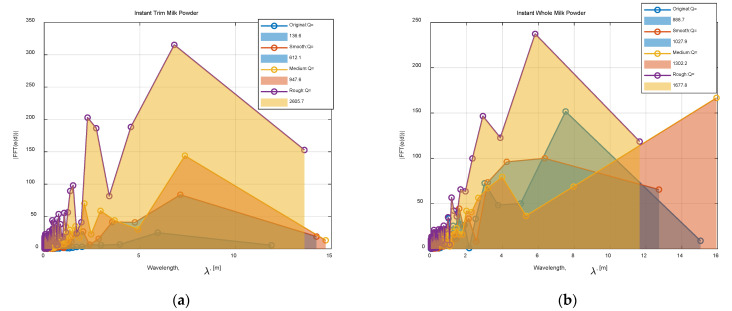
(**a**) Comparing the frequency response of the Class 0–3 instant trim samples; (**b**) comparing the frequency response of the Class 0–3 instant whole samples.

**Figure 15 foods-12-00967-f015:**
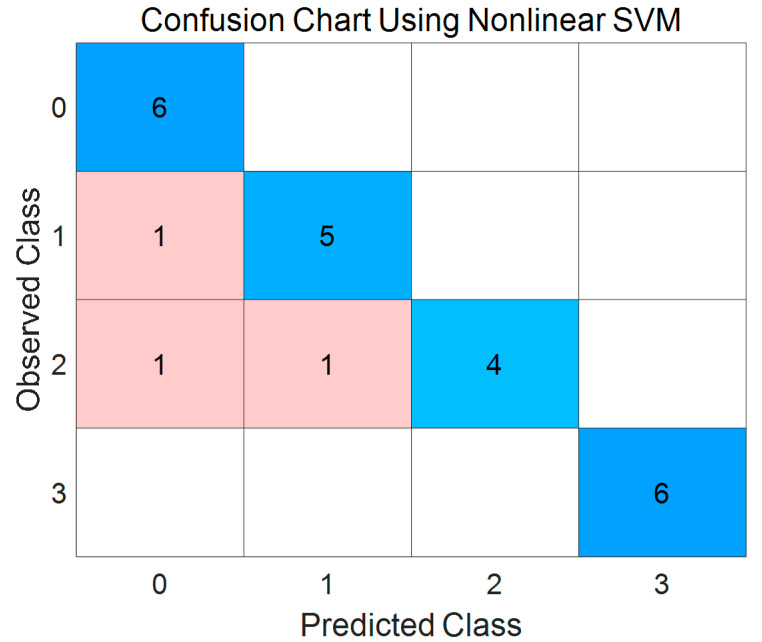
The confusion chart for the developed SVM classifier.

**Table 1 foods-12-00967-t001:** The actual moisture level.

Milk Powder Type	Class 0	Class 1	Class 2	Class 3
Instant Trim Milk Powder	5.91 ± 0.27%	8.19 ± 0.45%	9.79 ± 0.66%	11.98 ± 0.96%
Instant Whole Milk Powder	5.18 ± 0.23%	7.74 ± 0.51%	9.44 ± 0.64%	11.02 ± 0.87%

**Table 2 foods-12-00967-t002:** The specificities and sensitivities of the developed SVM model.

Class	Sensitivity	Specificity
Class 0	100%	75%
Class 1	83.3%	83.3%
Class 2	66.7%	100%
Class 3	100%	100%

## Data Availability

All related data and methods are presented in this paper. Additional inquiries should be addressed to the corresponding author.

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
