# Peer review of "Application of Three-Dimensional Digital Photogrammetry to Quantify the Surface Roughness of Milk Powder"

_foods, 2023, doi:10.3390/foods12050967_

Round 1

Reviewer 1 Report

The paper submitted to the journal Foods aims to develop an application based on 3D Digital Photogrammetry to Quantify the Surface Roughness of Milk Powder. The paper is very well written and didactic. Upon reading the paper, I recommend some modifications cited below.

Line 54. Please precise that this has been performed on "images".

Line 212. please explain why this number of contour and how it has been defined?

The end of the sentence is not clear. Why the lumps will have similar size? It is evident in the different photos presented in the paper that lumps have different size. Please explain.

Line 239. Please precise the number of the equation. Here equation number 1.

Please cite the equation in the texte line (Line 253).

Table 2 and figure 15. I think that the authors should present the results of calibration and Cross validation in order to have a clear idea of the model performance.

Reviewer 2 Report

Dear Editor and Authors,

I send you my review about the article “Application of 3D digital photogrammetry to quantify the surface roughness of milk powder”.

The scope of the paper, as reported in the aim was to develop a three dimensional digital photogrammetry technique for quantifying the surface roughness of commercial milk powders.

In my opinion, the article result sufficiently original, and enough well structured.

However, it needs some minor revisions that I report below.

In general, acronyms can help the reader better understand the article. However, their use should be avoid in the title.

Therefore, to facilitate the reading and understanding of the title of this article, I suggest that the authors to replace “3D” with “three dimension” in the title and to explicit that “3D” is the acronyms of “three dimension” the first time that it were use in the text.

The introduction result complete and well support the aim of the research and, although it is well written, it result too long and dispersive, thus it should be summarise.

Moreover, at line 55 were cited seven article, from reference 16 to 22, to support the phrase. I think that some of these reference should be delete.

Furthermore, the paragraph of materials and methods could result, too long and some part of it should need to be summarised, as for example, the paragraph of sample preparation.

In addition the paragraph "milk pownder samples" needs to be improved adding the numbers of the trials and of the replicate of analysis made.

The results is sufficiently well presented but they are not sufficiently discussed.

In particular the comparison of data shown with the one reported in literature is lacking.

Finally, the conclusions should be summarised and the text from line 379 to line 387 should be delete since it is a summary of what has just been read.

Reviewer 3 Report

The paper "Application of 3D Digital Photogrammetry to Quantify the Surface Roughness of Milk Powder" presents a novel method for characterization of surface roughness of milk powder and it is a valuable contribution to the knowledge of milk powders, and, therefore, suitable for publication in Foods. However, it is my opinion that the paper requires restructuring - especially in the Materials and Methods section which is too long and contains data which should be presented in the Results and discussion area. Furthermore, the discussion on some images is very short and needs to be supplemented with a more detailed insight into what those images represent, their specificity and their comparison to the state of the are. Please see the comments below:

Abstract

P1, L21: Remove the word "by" from the sentence.

Materials and Methods

The whole Materials and methods section should be restructured. Namely, it is my opinion that a large part of the text does not belong in the materials and methods, but in the Results section. E.g., the table with moisture contents is a result, since the milk powders were conditioned by "hand". Also, how was the moisture content determined. Please describe the method used. Furthermore, Fig.1 also belongs to the result and discussion section, especially since you  discussed the appearance of the powders in the figure in regard to their appearance which is connected to the moisture content. The same is valid for Figure 5 and Figure 7.

Results and discussion

P8, L283-287: Figure 9 should be discussed in more detail.

Figures 10 and 11 should also be discussed in more detail.

Reviewer 4 Report

As a reviewer, I would suggest that the authors of the manuscript entitled "Application of 3D Digital Photogrammetry to Quantify the Surface Roughness of Milk Powder" address the similarities between this manuscript and their previously published work "Assessing and Quantifying the Surface Texture of Milk Powder Using Image Processing" (https://doi.org/10.3390/foods11101519). The authors should explain what is different and new in this study and how it adds to the existing literature on the topic, as well as provide a detailed explanation of the methodology used in their study, and how it differs from their previous work. They should also clearly state the significance of their findings and how they contribute to the field. Additionally, it is important for the authors to acknowledge and address the fact that the same photographs of samples were used in both papers, which calls into question the originality and validity of the research presented.

The authors should demonstrate how this new study is distinct and makes a unique contribution, and clearly explain how their methodology differs from their previous work, and how it differs from other studies in the field. They should also provide a detailed explanation of how the proposed method of quantifying the surface roughness of milk powder samples differs from their previous work, which used surface normal analysis to quantify surface texture. Furthermore, it is important for the authors to acknowledge that simply reusing the same photographs and using a similar approach is not acceptable, as it violates ethical standards in academic research and is considered self-plagiarism.

In summary, as a reviewer, I would suggest that the authors address the similarities between their current manuscript and their previously published work, and provide a clear and detailed explanation of the differences and unique contributions of their current study. They should also address the use of the same photographs in both papers and provide a clear justification for why they were used in both studies. If no proper explanation or justification can be provided, it may call into question the integrity of the research presented in the manuscript. It is crucial for the authors to adhere to ethical standards in academic research and ensure the originality and transparency of their findings.

Round 2

Reviewer 4 Report

Thank you for making the suggested corrections to your manuscript. However, after re-reviewing the manuscript, I have noticed that several important issues still need to be addressed. 

-          It is imperative that you provide a clear introduction to the fact that this work is based on previous research and highlight the differences between the two in terms of the goal, methodology, and expected results. Additionally, the methodology used in your study should be clearly explained and compared to your previous work. I kindly request that you clarify this and make sure it is included in the M&M section.

-          Regarding the plagiarism claims, I understand that the method of preparing milk powder samples in this study is the same as in your previous work. However, it is important to provide a detailed explanation of the methodology in the Materials and Methods chapter and clarify what is different from the previous work. If the methodology is indeed the same as in your previous work, it would be important to clearly state this in the manuscript.

-          However, it would also be in line with originality and ethical considerations to provide new photos of the samples to support the validity and originality of the research presented in this manuscript. Using completely new photos of the samples will help to demonstrate originality and ethical standards in your research.

-          Finally, the corrections made so far have only been "cosmetic changes" and further concrete and extensive corrections are still required. I kindly request that you address these issues in order to improve the quality and validity of your research.

 Thank you for your attention to these matters. I look forward to your response.
